# How do multimedia and blended learning enhance music elective courses? examining the roles of learning attitudes, styles, and teaching presence

Pingbo Tang[1], Xiaoshi Zhou[2], Yuefeng Zhou[3], Rong Cheng[4]*

1 Department of Humanities and Music, Hunan Vocational College of Science and Technology, Changsha, Hunan, China, 2 College of Music, Hunan Normal University, Changsha, Hunan, China, 3 College of Music, Hunan Normal University, Changsha, Hunan, China, 4 Department of Arts, International Business Vocational College, Changsha, Hunan, China

* rc2116@columbia.edu

## Abstract

This study aims to explore the possibility of using Internet resources to enhance the educational effect of music elective courses in colleges and universities. Moreover, this study analysed students' perception of blended learning mode and their continuous learning intention. The study adopted the Technology Acceptance Model (TAM) and the theory of sustained learning intention as theoretical frameworks, combined with factors such as learning attitude and learning style, to explore the impact of these factors on students' learning outcomes. The study found that learning attitudes and styles are positively correlated with students continuous learning intention via questionnaire surveys and quantitative analysis, supporting the research hypothesis. The research results are of great significance for optimizing the design and teaching methods of music elective courses, providing theoretical support and empirical basis for promoting innovation in music education.

## 1. Introduction

The advent of the Internet has ushered in novel opportunities across diverse fields and has seamlessly integrated into the fabric of contemporary student life [1]. Serving as a pivotal conduit for information acquisition, it also functions as a dynamic platform for personal expression and intellectual discours [2]. Within this digital landscape, the fusion of college music elective courses with the Internet presents both unprecedented advantages and challenges to traditional music education. In alignment with the directives issued by the guidelines of the Chinese Ministry of Education concerning aesthetic education in higher education institutions, a multitude of teaching elective courses, encompassing art, sports, dance, among others, have been introduced. Notably, music elective courses in universities have consistently garnered significant

**Data availability statement:** All relevant data are within the manuscript and its Supporting information files.

**Funding:** The author(s) received no specific funding for this work.

**Competing interests:** The authors have declared that no competing interests exist.

attention from students due to their inherent allure. According to the 2022 National College Students' Elective Course Survey Report, art elective courses, with music being a prominent component, emerged as the most preferred option among college students, accounting for 48.3%, of respondents. Conversely, economics and management courses received the least preference, comprising merely 7.1% of selections. This data underscores not only the popularity of music courses among students but also their vital role in fostering comprehensive quality education. Music electives contribute to nurturing students' musical literacy, augmenting their aesthetic appreciation, and providing an avenue for emotional expression and stress relief [3,4]. Online platforms facilitate seamless access to an array of music resources, enabling students to delve into the narratives embedded within musical compositions and even engage in dialogues with esteemed musicians [5]. This hybrid learning paradigm significantly enriches students' musical experiences and paves the way for innovative teaching methodologies in music elective courses, as illustrated in Fig 1.

In the rapid-paced modern society, sustaining students' interest in music elective courses has emerged as an intriguing research domain meriting further exploration [6]. Maintaining such interest not only amplifies students' learning motivation but also positively impacts their mental wellbeing [7]. Moreover, Music learning activities may promote cognitive development and improve students' emotional expression abilities [8]. Thus, the exploration of strategies to ignite and sustain students' enthusiasm for music electives in the digital era presents a compelling research agenda.

The concept of learning intention (LI) has been studied extensively over the past decades across both academic and practical fields [9]. LI pertains to an individual's willingness or readiness to engage in a specific learning activity, shaped by various factors such as motivation, the perceived relevance of content, and personal learning goals. In the realm of music education, LI holds particular significance as it influences not only students' choices in elective courses but also their overall engagement and persistence in music learning [10]. Recent years have witnessed innovations in the teaching of music elective courses, primarily driven by the Internet and novel teaching methodologies, aimed at effectively catering to students' physical and mental developmental needs [11]. These innovations manifest in various

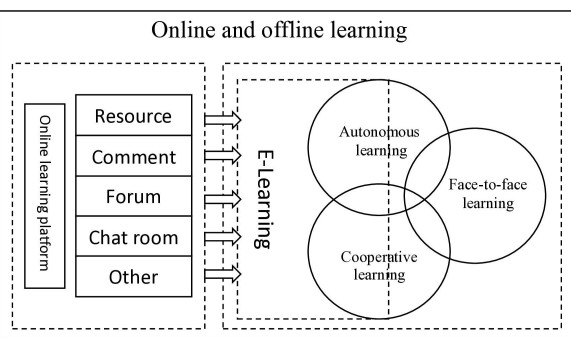

**Fig 1. Blended Learning Design.**

forms, including online learning platforms, interactive learning tools, and virtual reality technology, which not only provide abundant learning resources but also facilitate real-time interaction, thereby fostering collaboration and communication among students. Despite numerous studies demonstrating the positive impact of specific elective courses on students' performance across various subject areas and evaluating student satisfaction at the conclusion of courses [12], there remains a notable research gap concerning students' expectations and continuous learning intention prior to enrolling in music elective courses. This gap necessitates addressing through a comprehensive literature review, highlighting a crucial research opportunity within the field of music education. Furthermore, the preference of college students for teaching methods in music elective courses and their acceptance of diverse pedagogical approaches in music education have not been systematically evaluated [13]. Such an evaluation would be instrumental in enabling educators to optimize course design to better align with students' needs. To bridge the knowledge gap, this study investigated the impact of social presence, cognitive presence, and teaching presence on students to reflect the vividness of student course interaction, which reflects the effectiveness of the course through students' own perceptions. Additionally, this study contributes to the existing literature by exploring the role of perceived playfulness, which encapsulates the charm and joy students experience in music elective courses, thereby enhancing their learning satisfaction and confidence [14] This research broadens our understanding of perceived course quality, engaging course design, and effective strategies for teachers to capture students' attention, stimulate their curiosity, and nurture their thirst for knowledge. Such long-term interest may ultimately contribute to better academic performance in music learning and encourage students to delve deeper into the field of music in the future.

With the rapid development of information technology, the education sector has undergone unprecedented transformations. Music, an indispensable component of human culture, has similarly evolved in its teaching methodologies [15]. Nicolaou et al. (2019) argue that the application of multimedia technology in classroom teaching provides a robust material foundation for enhancing educational quality in schools [16]. In the context of music elective courses, integrating and optimizing diverse teaching resources through the blended learning model offers students more flexible and diverse learning methods, along with abundant learning resources. This model not only addresses individual learning needs but also augments teachers' instructional efficiency [17]. To effectively merge traditional music theory with practical performance, teachers can devise varied teaching activities, leveraging online platforms for course discussions where students can share their performance experiences and feelings, thereby reinforcing theoretical knowledge and enhancing practical skills [18]. Furthermore, online platforms facilitate activities such as discussions and interactions, which stimulate students' interest and enthusiasm for learning. From the theoretical perspective, the findings offered a novel perspective that blended learning promotes students' intentions to learn music elective courses. This is still an unexplored area in current literature [4]. This study aims to elaborate on the relationship among these factors, considering perceived usefulness and student satisfaction as mediating variables and learning style as a moderating factor. By doing so, it seeks to provide a more comprehensive and dynamic understanding of how these variables interact and influence students continued learning intentions in music education.

## 2. Theoretical background

### 2.1. Concept of continuous learning intention

Learning intention is the degree to which an individual judges their tendency to engage in a certain learning behavior, derived from Bhattacharjee's concept of sustained intention [19,20]. He emphasized that sustained intention or behavior refers to the user's sustained intention or behavior towards a certain technology or thing over a longer period. Within this theoretical framework, learning intention transcends mere short-term behavioral reflection, embodying a dedication to long-term learning continuity. Consequently, understanding learning intention is crucial for the design of effective educational system [21]. The research on the continuous use of information systems has aroused the interest of many scholars and has been successfully applied to various information systems such as learning platforms, social network services,

and mobile applications. Online course design based on interactive perspectives can enhance learners' social interaction, reduce anxiety and loneliness in online learning, increase learners' motivation, and thus enhance their continuous learning intention [22]. Kingsford-Smith (2021) believed that the continuous learning intention is the behavioral willingness of learners to complete current course tasks [23]. Elpus(2022) proposed incorporating gamification elements into course design to enhance learners' interest in learning and strengthen online user interaction; Teachers pay attention to interaction and communication among students during the teaching process to enhance learners' continuous learning intention [24]. Moreover, Yao et al. (2023) explored the strategies to be considered in the design of blended courses from the perspective of improving students' autonomous learning ability when studying their learning attitudes and sustained intention in blended courses [25].

### 2.2  Theories of the technology acceptance model

The Technology Acceptance Model (TAM), proposed by American scholar Davis using rational behavior theory, is a framework designed to investigate user acceptance of information systems. This model encompasses two primary aspects: (1) perceived usefulness, where individuals assess whether a technology is beneficial to them based on their personal knowledge, experience, and recommendations from others; and (2) perceived ease of use, where individuals evaluate their satisfaction with a new technology based on their existing knowledge and skills [26,27] The TAM's central focus is to understand how users adopt and utilize new technologies, which is particularly crucial in educational environments for enhancing learning efficiency. To delve deeper into the dynamic process of technology acceptance, the Expectation-Confirmation Model of Information Systems Continuance (ECM-ISC), reconstructed by Bhattacherjee was introduced. The ECM-ISC comprises four key variables: expected confirmation, satisfaction, intention to continue using, and perceived usefulness. This model emphasizes the relationship between expectation confirmation and satisfaction during the technology use process. Specifically, expectation confirmation pertains to users' anticipations regarding the functionality and effectiveness of the technology, whereas satisfaction is the evaluation of these expectations after actual utilization. These factors further influence users' intention to continue employing the technology, thereby augmenting the core components of the TAM model. In this context, perceived usefulness and satisfaction are recognized as significant predictive factors and play a mediating role. The theory of the technology acceptance model is predominantly applied in various fields, including network technology, online learning, mobile learning, and blended learning [28]. For instance, Dagdeler et al. (2020) examined the impact of Mobile-Assisted Language Learning (MALL) on learners in English education [29]. Lei (2022) incorporated concepts of intrinsic motivation and psychological construction into the technology acceptance model, demonstrating its explanatory power in the context of MALL. In sum, this study is based on the TAM model and integrates relevant variables such as perceived interest and perceived quality to reflect students' actual learning motivation in music classes [30].

## 3.  Hypotheses

### 3.1  Conceptual model

The conceptual model is shown as Fig 2, examines the relationships between various factors influencing students' continuous learning intention. Key constructs include perceived playfulness, cognitive presence, teaching presence, social presence, and perceived quality, all of which impact perceived usefulness and satisfaction. These in turn affect learning attitude, learning style, and ultimately, continuous learning intention. This model provides a comprehensive framework for testing hypotheses about how these constructs interact within a blended learning environment in music education.

### 3.2  Learning attitude

Learning attitude is defined as a relatively stable psychological tendency exhibited by students towards learning situations, which is the positive or negative response tendency formed by students towards learning [31]. This emphasizes the

importance of learning attitude in students' learning process, especially how it affects their learning experience and outcomes. Compared to learning attitude, learning motivation is the intrinsic force that drives students to engage in learning activities. Learning motivation typically includes interest in learning, expectations for learning goals, and students' confidence in their own abilities [32]. Therefore, although these two concepts are interconnected to some extent, learning attitude reflects more of the emotional response students have towards the learning process. In contrast, learning motivation is closely related to behavioral drive. This suggests that students' learning motivation directly affects their willingness to learn. According to Clauhs (2022), learning motivation influences online learning satisfaction and learning attitude, which in turn affects students' continuous learning intention [33]. For example, there is a significant positive correlation between the positivity of learning attitude and learning motivation [34]. When students hold a positive attitude towards learning, their learning motivation is usually high, and this high level of motivation helps increase their willingness for continuous learning [35]. Thus, clarifying the distinction and connection between these two concepts is crucial for understanding student learning behavior. Hence, the following hypothesis is constructed.

H1: Learning attitude positively correlates with continuous learning intention.

## 3.3 Learning style

Learning style refers to the relatively stable cognitive thinking patterns formed by students during the long-term learning process, which enables them to adopt personalized thinking methods to solve problems. Pendergast determined that learners with visual learning styles perform better than other types of learners [36]. Culp (2021) also found a certain relationship between students' learning style preferences and academic performance [37]. Chen (2022) focused on the relationship between learning styles and music learning, as well as the promoting effects of each learning style on music learning. Hence, the hypotheses formulated as follows [38].

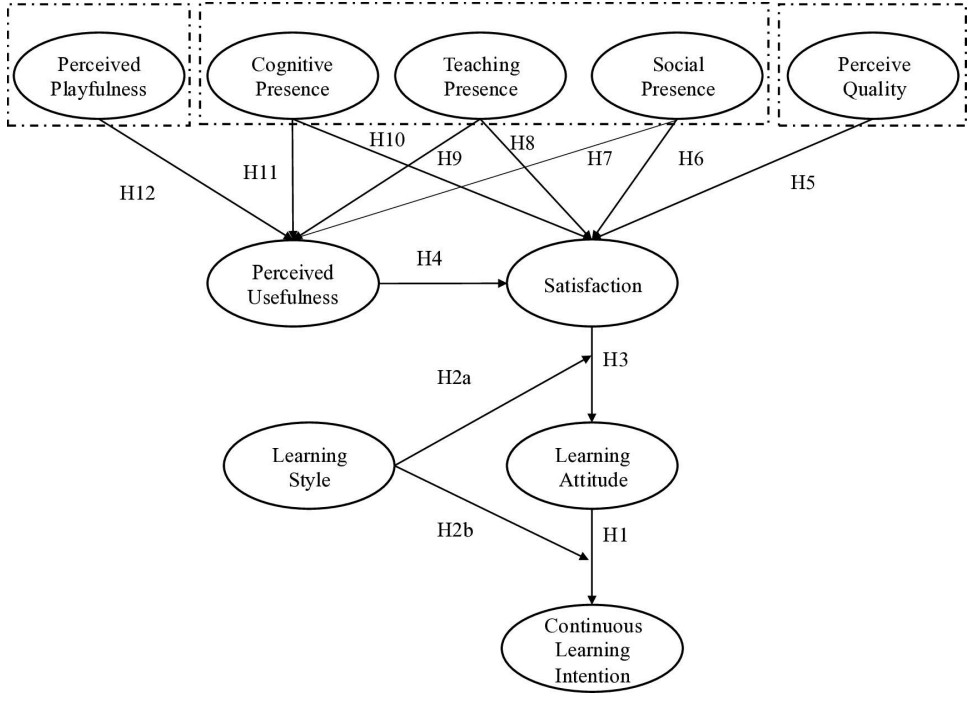

**Fig 2. Research Model.**

H2a: Learning style moderates the relationship between perceived usefulness and learning attitude.

H2b: Learning style moderates the relationship between learning attitude and continuous learning intention.

### 3.4 Satisfaction

Student satisfaction is one of the important indicators used to evaluate the effectiveness of course learning [39]. Especially in the field of remote learning, satisfaction refers to the degree of satisfaction that online learners have with their learning outcomes. Kingsford-Smith et al. (2021) defined learning satisfaction as the perception of pleasure and achievement in the learning environment. In the learning environment, the relationship between satisfaction and learning attitude has received increasing attention [40]. When students feel satisfied with their learning experience, their learning attitude tends to become more positive. High satisfaction typically enhances students' intrinsic motivation, encouraging them to engage more willingly in learning activities and face challenges in learning with a positive outlook [41]. Furthermore, satisfaction is closely related to the willingness to continue learning. This means that in a positive learning experience, students not only feel joy but also develop a more positive attitude towards future learning [42]. Therefore, satisfaction is considered an important factor influencing learning attitude. Hence, the hypothesis is formulated as follows.

H3: Student satisfaction positively correlates with learning attitude.

### 3.5 Perceived usefulness

The concept of perceived usefulness originates from Davis' TAM model, which refers to the extent to which users believe that new information technologies can help them perform or work better. In the blended learning, perceived usefulness is the degree to which students evaluate the optimization of their learning process, improvement of learning efficiency, and optimization of learning performance with the support of technology. It is closely related to learning satisfaction and willingness to continue learning. Research has shown that perceived usefulness can affect users' attitudes towards a certain behavior [43]; The perceived usefulness of adult learners participating in distance learning has a significant impact on their satisfaction [44]; Blom (2017) found that the perceived usefulness of mobile Internet services has a significant impact on adult user satisfaction. Hence, the following hypothesis is constructed [1].

H4: Perceived usefulness positively correlates with student satisfaction.

### 3.6 Perceived quality

The perceived quality of a course is a key factor in measuring its success or failure [45]. The perceived quality of a course is defined as the overall impression and feeling of learners towards the course, including teaching content, teaching methods, teaching facilities, teacher-student interaction, and other aspects [46]. Perceived quality is not only influenced by the course design itself, but also by various factors such as the learner's personal background and learning environment [47]. Zhao et al. found that perceived quality has a strong predictive effect on learning satisfaction and willingness to continue learning in courses [48]. Thus, the hypothesis formulated as follows.

H5: Perceive quality positively correlates with student satisfaction.

### 3.7 Social presence

Social presence is the social experience of students in the learning environment. Social presence provides students with a good online learning experience from the perspective of emotional expression and communication [49]. Social presence reflects the degree to which learners have social and emotional connections with others in the online environment, and learners should have the ability to express their emotions and socialize [50]. The role of social existence is to support cognitive and emotional goals of learning. When participants feel happy and satisfied in interactions, they will maintain a continuous state of participation, promoting critical thinking among learners [51]. Scholars have explored and confirmed

that social presence has a significant impact on users' cognition, attitudes, and behavioral intentions in various online environments. Onderdijk proposed that there is a significant correlation between the level of social presence and learner satisfaction [52]; Song et al. (2020) believed that learners' learning experience is positively influenced by social presence [53]; Huqiu et al., (2020) confirmed that social presence is an important predictor of online learner satisfaction. In addition, the satisfaction and sense of belonging of SNS users serve as intermediaries between their willingness to continue using social presence [54]. Thus, the hypotheses formulated as follows.

   H6: Social presence positively correlates with student satisfaction.

   H7: Social presence positively correlates with perceived usefulness.

### 3.8 Teaching presence

Teaching presence refers to students' perception of the existence of teachers, this includes the teacher's physical presence, the transmission of professional knowledge, and the level of active involvement of the teacher in the learning process. Teachers enhance this sense of teaching presence by designing clear learning activities, organizing coherent teaching content, and actively responding to students' questions and needs. This presence is crucial to the success of a learning community, Teaching presence is the key to the success of a learning community, helping students gain a good online learning experience [55]; Thus, this study considers the sense of teaching presence as an important criterion for students to evaluate their blended learning experience. The manifestation of teaching presence mainly lies in its teaching design and organization, as well as teaching behavior, including designing, or organizing clear and explicit learning activities for students, providing corresponding learning resources, and timely feedback to students. These behaviors are all manifestations of teaching presence [56]. Bremmer (2022) proposed that there is a significant correlation between learners' level of teaching presence and their satisfaction; Hrabluk (2023) supported that the sense of teaching presence has the most important impact on learners' perception of learning and satisfaction [57]. Therefore, we can construct the following hypotheses to explore the impact of teaching presence on students' learning experience and satisfaction in more depth. Hence, the following hypotheses are constructed.

   H8: Teaching presence positively correlates with student satisfaction.

   H9: Teaching presence positively correlates with perceived usefulness.

### 3.9 Cognitive presence

Cognitive presence is defined as the degree to which students engage in sustained and critical self-reflection within the curriculum [58] Cognitive presence can help students achieve a good online learning experience. Research has shown that cognitive presence can help improve students' satisfaction with their learning experience [59]; There is a certain correlation between the level of cognitive presence and learner satisfaction [60]; Satterfield (2023) et al. found that cognitive presence has the most significant impact on the effectiveness of online learning. Hence, the following hypotheses are constructed [61].

   H10: Cognitive presence positively correlates with student satisfaction.

   H11: Cognitive presence positively correlates with perceived usefulness.

### 3.10 Perceived playfulness

Perceived playfulness (PP) refers to the state in which an individual participates in an activity because it allows them to feel pleasure and joy and is the state in which they interact with a situation [62]. Multiple studies have shown that perceived playfulness has a significant impact on perceived usefulness [63]. Balkaya (2021) suggests that perceived interest has a significant positive predictive effect on performance expectations. There is a significant positive predictive effect between perceived playfulness and perceived usefulness [64]. In addition, Clifford's (2024) study on the influencing

factors of college students' intention to use mobile learning showed a significant positive correlation between perceived playfulness and perceived usefulness [65]. Therefore, the following assumptions were established.

H12: Perceived playfulness positively correlates with perceived usefulness

## 4. Experimental results

### 4.1 Source of questionnaire questions

The questionnaire designed in this study aims to comprehensively explore the relationship between students' learning experience and academic achievement. The questionnaire is divided into three parts, considering factors such as learning experience, social interaction, and technological implementation, to comprehensively evaluate students' academic participation and individual achievements. First, this section covers multiple key factors, including intention to continue learning, learning attitude, learning style, student satisfaction, perceived usefulness, perceived course quality, etc. These factors not only directly affect students' academic performance and participation, but also profoundly impact their overall perception and evaluation of the learning process. Second, a partial assessment was conducted on students' level of interaction with peers and teachers during the learning process, as well as their perception of teaching quality and academic challenges, including social presence, teaching presence, and cognitive presence. The third part focuses on the measurement of technology implementation and interest, especially the impact of perceived interest on students' learning motivation and interactivity. The design of this questionnaire draws on previous research results to ensure the measurement tool more scientific and reliable. The detailed measurement items are shown in Table 1, and the reliability and validity of the questionnaire are ensured through extensive surveys and data analysis. This survey was conducted from January 12, 2024, to May 29, 2024, at Hunan University of Science and Technology. A stratified sampling method was used to target first-year, second-year, third year, and fourth-year students at the university. We distributed 400 questionnaires. Half of these (200) were distributed online, while the other half (200) were distributed offline. In the end, we collected 315 valid questionnaires. Data collection was done anonymously to ensure students' privacy and the authenticity of the data. When collecting data, we also ensured that students had enough time and space to carefully think about each question. This helped improve the validity and reliability of the questionnaire. The sources of the measurement items adapted from prior studies are displayed in Table 1.

### 4.2 Sample

This study analyzed the basic information of participants in detail through a questionnaire survey, including age, gender, grade, and major, as shown in Fig 3. The sample mainly consists of students under the age of 18 (51.4%), followed by

**Table 1. Sources of measurement items in the questionnaire.**

| Constructs | Number of Measurement Items | Sources |
|---|---|---|
| Continuous Learning Intention | 5 | Nicolaou, Constantinos et al.(2019); Dagdeler, Kubra Okumus et al.(2020) |
| Learning Attitude | | Dagdeler, Kubra Okumus et al.(2020); Lei, Xiao et al. (2022) |
| Learning Style | 4 | Culp, Mara E et al. (2021) ; Shaw et al. (2022) |
| Student Satisfaction | 4 | Freer et al. (2019); Kingsford-Smith et al. (2021) |
| Perceived Usefulness | 4 | Leong et al. (2011); Bautista et al. (2022) |
| Perceive Quality | 6 | Chen, et al., (2019); Melzner et al., (2023) |
| Social Presence | 5 | Cyr, et al., (2007); Scorolli et al., (2023) |
| Teaching Presence | 5 | Hibbard, et al., (2017); Hrabluk et al.,(2023) |
| Cognitive Presence | 5 | (Hrabluk, Robert, 2021); Satterfield (2023) |
| Perceived Playfulness | 4 | Wang et al., (2022); Clifford et al., (2024) |

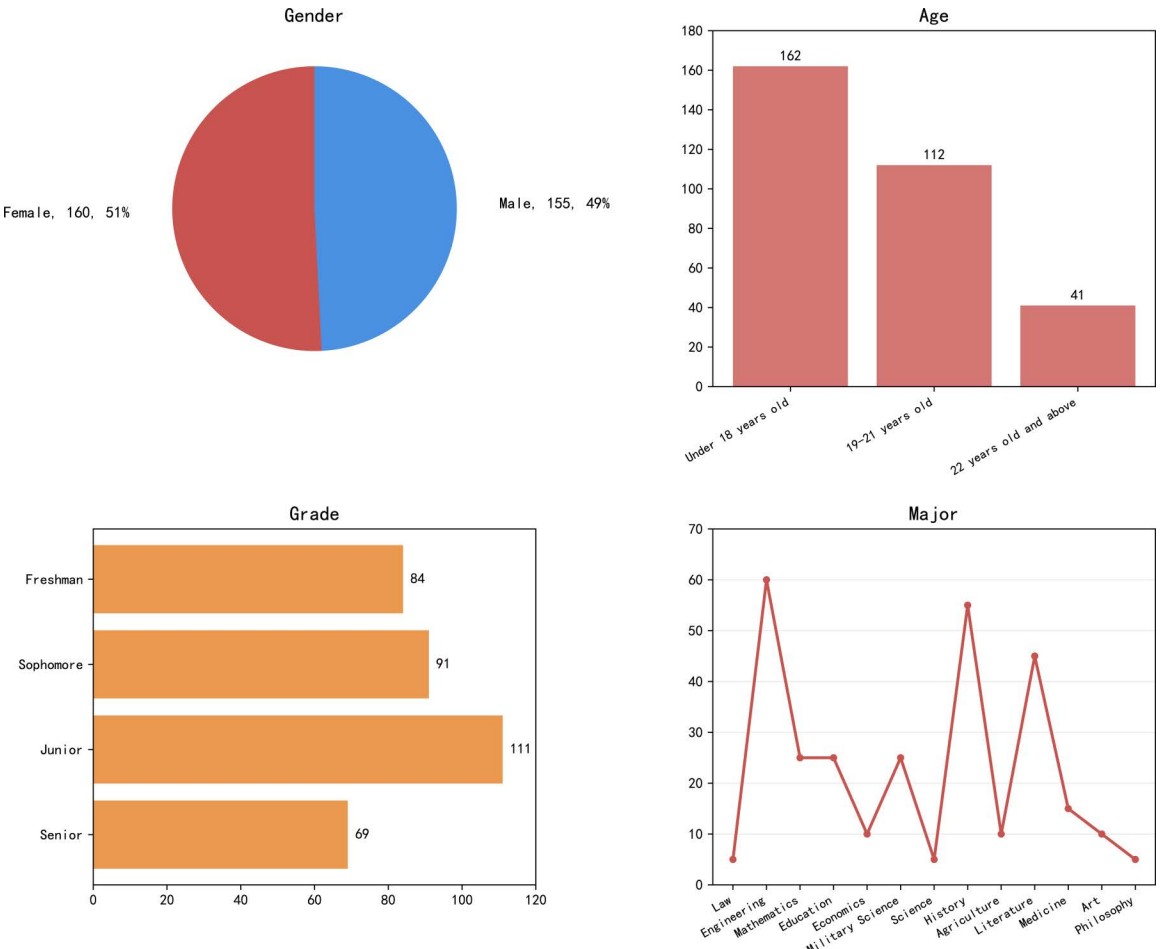

**Fig 3. Basic information analysis.**

students aged 19–21 (35.6%), with a relatively small proportion of students aged 22 and above (13%). This indicates that the participants are mainly junior college students, which meets the research design objectives. Gender distribution: The gender ratio is close to equilibrium, with slightly more girls than boys (51%: 49%). This balanced distribution helps to reduce the impact of gender bias on research results. In addition, the sample covers students from grades one to four, and this distribution reflects the learning experience and academic achievement of students in different grades. The participants come from multiple majors, with engineering majors being the main ones (19%), followed by management (14.6%) and science majors (13.5%). There are relatively few participants in majors such as law, military science, agriculture, philosophy, and art, accounting for 1.6% to 2.9%. This diversified professional distribution provides a comprehensive perspective for analyzing the differences in students' learning experiences under different disciplinary backgrounds.

### 4.3. Exploratory Factor Analysis (EFA) Results

**1) Reliability and validity analysis.** This study evaluated the internal consistency of the measurement tool by calculating Cronbach's alpha coefficient. Table 2 shows that the overall reliability of the measurement tool is high, with a Cronbach's alpha coefficient of 0.929. When Cronbach's alpha coefficient is greater than 0.7, the measurement tool has good internal consistency, and the stability and consistency of its measurement results can be trusted. Therefore,

**Table 2. Reliability and validity.**

| Item | Alpha | Number of Items | KMO Sampling suitability quantity | Bartlett's test of sphericity | | |
| --- | --- | --- | --- | --- | --- | --- |
| | | | | Approx. Chi-Square | df | p-value |
| Value | .929 | 46 | .901 | 8759.316 | 1035 | .000 |

the measurement tool used in this study demonstrated high reliability in the sample, effectively ensuring the reliability and consistency of the data. To evaluate the effectiveness of the measurement tool, Kaiser Meyer Olkin (KMO) measurement and differential item analysis were used. The KMO measurement value is 0.901, which is much higher than the recommended threshold of 0.6, indicating that the sample has good suitability and is suitable for factor analysis. Furthermore, according to the results of the approximate chi square test, the sample data showed a significant correlation structure during factor analysis ($\chi^2 = 8759.316$, df = 1035, $p < .001$) 。 These results indicate that the measurement tool designed in this study has good effectiveness in structure and internal measurement and can effectively capture the complex relationships and patterns between research variables.

2) **Factor analysis.** This study explored the potential factor structure in measurement tools using principal component analysis (PCA) and variance maximization rotation methods. As shown in Table 3, firstly, the cumulative variance percentage of the first five principal components reached 52.873%, indicating a high explanatory power of the latent factors in the measurement tool. This indicates that these principal components can effectively explain the variability observed in measurement tools and provide important information for studying the complex relationships of variables. Secondly, from the sixth to the tenth principal components, although the variance contribution of each principal component is relatively low, their cumulative variance percentage is still significant, reaching 71.321% of the overall population. This indicates that in principal component analysis, selecting the top ten principal components can effectively capture most of the data variability, which helps to deepen the understanding of the multidimensional structure of measurement tools. Therefore, through the analysis of the number of factors, we have confirmed that selecting ten principal components as the appropriate number for factor analysis. These principal components not only have good explanatory power, but also maintain the relative simplicity of the data, making subsequent factor rotation and structural equation modeling analysis more effective and reliable.

This study used principal component analysis (PCA) combined with variance maximization rotation (Varimax) technique to explore the factor structure of the measurement tool in depth. The rotated component matrix clearly reveals that each questionnaire item is mainly loaded on a single factor and has a high loading coefficient, presenting a concise and

**Table 3. Results of Factor Analysis.**

**Explanation of Total Variance**

| Component | Initial eigenvalue | | | Sum of squared rotational loads | | |
| --- | --- | --- | --- | --- | --- | --- |
| | Total | Variance percentage | % | Total | Variance percentage | % |
| 1 | 11.986 | 26.057 | 26.057 | 4.349 | 9.454 | 9.454 |
| 2 | 3.951 | 8.589 | 34.646 | 3.831 | 8.329 | 17.783 |
| 3 | 3.153 | 6.854 | 41.500 | 3.551 | 7.720 | 25.503 |
| 4 | 2.841 | 6.176 | 47.676 | 3.502 | 7.613 | 33.115 |
| 5 | 2.391 | 5.197 | 52.873 | 3.492 | 7.591 | 40.706 |
| 6 | 2.189 | 4.758 | 57.631 | 2.963 | 6.442 | 47.148 |
| 7 | 2.033 | 4.420 | 62.052 | 2.915 | 6.337 | 53.484 |
| 8 | 1.701 | 3.698 | 65.750 | 2.877 | 6.255 | 59.739 |
| 9 | 1.409 | 3.062 | 68.812 | 2.721 | 5.914 | 65.653 |
| 10 | 1.154 | 2.509 | 71.321 | 2.607 | 5.668 | 71.321 |

**Table 4. Rotated component matrix[a].**

| | Component | | | | | | | | | |
|---|---|---|---|---|---|---|---|---|---|---|
| | 1 | 2 | 3 | 4 | 5 | 6 | 7 | 8 | 9 | 10 |
| PCQ3 | .802 | | | | | | | | | |
| PCQ2 | .795 | | | | | | | | | |
| PCQ5 | .795 | | | | | | | | | |
| PCQ6 | .792 | | | | | | | | | |
| PCQ4 | .787 | | | | | | | | | |
| PCQ1 | .751 | | | | | | | | | |
| SP3 | | .880 | | | | | | | | |
| SP5 | | .877 | | | | | | | | |
| SP1 | | .861 | | | | | | | | |
| SP2 | | .857 | | | | | | | | |
| SP4 | | .855 | | | | | | | | |
| TP3 | | | .820 | | | | | | | |
| TP1 | | | .793 | | | | | | | |
| TP2 | | | .774 | | | | | | | |
| TP5 | | | .761 | | | | | | | |
| TP4 | | | .750 | | | | | | | |
| CP4 | | | | .782 | | | | | | |
| CP5 | | | | .775 | | | | | | |
| CP2 | | | | .774 | | | | | | |
| CP1 | | | | .766 | | | | | | |
| CP3 | | | | .741 | | | | | | |
| CLI2 | | | | | .816 | | | | | |
| CLI5 | | | | | .787 | | | | | |
| CLI3 | | | | | .786 | | | | | |
| CLI1 | | | | | .781 | | | | | |
| CLI4 | | | | | .765 | | | | | |
| LA2 | | | | | | .793 | | | | |
| LA3 | | | | | | .785 | | | | |
| LA4 | | | | | | .752 | | | | |
| LA1 | | | | | | .746 | | | | |
| SS1 | | | | | | | .791 | | | |
| SS4 | | | | | | | .789 | | | |
| SS3 | | | | | | | .773 | | | |
| SS2 | | | | | | | .761 | | | |
| PP4 | | | | | | | | .813 | | |
| PP2 | | | | | | | | .788 | | |
| PP3 | | | | | | | | .786 | | |
| PP1 | | | | | | | | .781 | | |
| PU1 | | | | | | | | | .796 | |
| PU4 | | | | | | | | | .761 | |
| PU2 | | | | | | | | | .728 | |
| PU3 | | | | | | | | | .694 | |
| LS2 | | | | | | | | | | .758 |
| LS1 | | | | | | | | | | .742 |
| LS4 | | | | | | | | | | .716 |

*(Continued)*

**Table 4.** (Continued)

| | Component | | | | | | | | | |
|---|---|---|---|---|---|---|---|---|---|---|
| | 1 | 2 | 3 | 4 | 5 | 6 | 7 | 8 | 9 | 10 |
| LS3 | | | | | | | | | | .666 |

Extraction method: Principal Component Analysis.

Rotation method: Caesar normalization maximum variance method.

The rotation has converged after 7 iterations.

Note. LA=Learning Attitude; LS=Learning Style; PU=Perceived Usefulness; PQ=Perceived Quality; SP=Social Presence; TP=Teaching Presence; CP=Cognitive Presence; PP=Perceived Playfulness.

clear factor structure. Specifically, the ten main concepts of perceived course quality, social presence, teaching presence, cognitive presence, continuous learning intention, learning attitude, student satisfaction, perceived interest, perceived usefulness, and learning style are composed of corresponding questionnaire items, with a load coefficient range of 0.666 to 0.880, demonstrating a high degree of concentration and explanatory power (as shown in the Table 4). Varimax rotation not only improves the discrimination between factors, ensuring the simplicity and reliability of factor structures, but also effectively avoids the problem of factor overlap. This factor analysis not only simplified the factor structure, but also maintained high explanatory power, laying a solid foundation for subsequent confirmatory factor analysis and structural equation modeling analysis.

### 4.4. Confirmatory Factor Analysis (CFA) Results

**1) Model indicator analysis.** This study conducted CFA using a dataset and obtained results. According to the validation factor analysis of the model fit index in Table 5, first, the CMIN/DF (chi square value divided by degrees of freedom) is 1.268, which is within the ideal range (less than 3), indicating a good fit of the model data. This suggests that the deviation between the observed data and the assumed structural model is small, and the model performs well in describing data variation. Second, according to indicators such as IFI, TLI, and CFI, all exceeded the standard value of 0.9, indicating that the model exhibits good fit under different fit indicators. Although NFI, GFI, and RFI are slightly lower than 0.9, they do not affect the overall fit of the model (Huang et al., 2013). Specifically, the values of IFI, TLI, and CFI all approached or exceeded 0.97, further confirming the excellent performance of the model in interpreting data. In addition, the GFI is 0.877, which also exceeds the standard value of 0.9, indicating a high overall fit of the model. This indicates that the model can adapt well to the observed data variability, providing an accurate description of the measurement tool structure. Finally, based on the RMSEA of 0.029, which is far below the standard value of 0.08, the good fit of the model was further verified. A low RMSEA value indicates that the model has small errors in data fitting and is suitable for explaining the observed data variability in measurement tools. In summary, we confirm that the confirmatory factor analysis model exhibits very good results in terms of fit. The excellent performance of these indicators provides a solid foundation and reliable support for further explanation and analysis of the structure of measurement tools and variable relationships in the future.

**2) Convergence validity analysis.** According to the convergence validity analysis results in Table 6, first, the average variance explained (AVE) of each variable is higher than 0.7, and even most of them are higher than 0.8, indicating that

**Table 5. Model Fit of Validation Factor Analysis.**

| Model Fit | CMIN | DF | CMIN/DF | NFI | RFI | IFI | TLI | CFI | GFI | RMSEA |
|---|---|---|---|---|---|---|---|---|---|---|
| Fit Results | 1012.22 | 798 | 1.268 | 0.877 | 0.867 | 0.971 | 0.969 | 0.971 | 0.877 | 0.029 |
| Judgment Std. | – | – | <3 | >0.9 | >0.9 | >0.9 | >0.9 | >0.9 | >0.9 | <0.08 |

**Table 6. Results of Convergence Validity Analysis.**

| Variable | Perceived Playfulness | Cognitive Presence | Teaching Presence | Social Presence | Perceive Course Quality | Perceived Usefulness | Student Satisfaction | Learning Attitude | Continuous Learning Intention |
|----------|----------------------|---------------------|-------------------|-----------------|------------------------|---------------------|---------------------|-------------------|-------------------------------|
| AVE | 0.85 | 0.78 | 0.81 | 0.79 | 0.83 | 0.77 | 0.80 | 0.82 | 0.84 |
| CR | 0.92 | 0.88 | 0.90 | 0.89 | 0.91 | 0.87 | 0.89 | 0.90 | 0.91 |

the various constructs of the measurement model explain a considerable amount of variance in the observed variables behind them. Specifically, the AVE of perceived playfulness reached 0.85, the highest among all variables, demonstrating its significance and stability in the measurement model. Second, the values of constructing internal consistency (CR) are all higher than 0.8, and most are even close to 0.9, indicating that the internal consistency of various variable constructs in the measurement model is good. For example, the CR of perceived playfulness is 0.92, indicating evidence of high correlation between the observed variables behind it. These indicators demonstrate that the measurement tool, supported by theoretical assumptions, can effectively capture the various concepts and their interrelationships of interest in the research. A high AVE value indicates that the measurement tools for each variable have high explanatory power at the conceptual level, while a high CR value further verifies the consistency and stability of the internal items of the measurement tools.

### 4.5 Structural Equation Modeling (SEM) results

In addition to EFA and CFA, this study conducted SEM analysis. Given the adequate sample size of this research, the data passed both the Kaiser-Meyer-Olkin (KMO) test (value = 0.901) and Bartlett's test of sphericity ($p < .001$), confirming the appropriateness of factor analysis. To ensure model fit, we adhered to rigorous criteria, with all fit indices (including CMIN/DF, CFI, TLI, and RMSEA) reaching desirable levels. While EFA, CFA, and SEM can effectively share the same dataset when data quality is high and sample size sufficient (Kyriazos, 2018), this study further validated result reliability through robustness tests including cross-validation. Consequently, we performed fitting and significance tests for SEM path analysis. The current research comprehensively demonstrates the rationality and effectiveness of employing rigorous analytical methods using the same dataset.

**1) Structural equation modeling path analysis.** According to the results of structural equation modeling path analysis in Table 7, this study can further evaluate the validity of each path in the research. Firstly, the path coefficient of perceived usefulness on perceived playfulness is 0.198, with a significance level ($p < 0.001$), supporting the hypothesis that perceived usefulness has a positive impact on perceived playfulness. This indicates that, in the context of research, increasing the perceived usefulness of users helps to enhance their perceived level of enjoyment. Second, the path coefficient of perceived usefulness on cognitive presence is 0.287, with a significance level ($p < 0.001$), which also supports the hypothesis that perceived usefulness has a positive impact on cognitive existence. This means that the perceived increase in system usefulness by users may promote their cognitive engagement in the educational environment. For teaching presence, the path coefficient of perceived usefulness is 0.240, with a significance level ($p < 0.001$), which also supports the hypothesis. This indicates that an increase in users' perception of system usefulness may enhance the level of perception of teaching existence, thereby improving the quality of teaching. However, the path coefficient of perceived usefulness for social presence is 0.073, with a significance level of 0.088, which did not reach the significance level ($p > 0.05$), and therefore cannot support this hypothesis. This means that in this study, users' perception of system usefulness did not significantly affect their perception of presence in social environments. In addition, the path coefficient between satisfaction and cognitive presence is 0.014, which does not reach a significant level ($p = 0.852$), therefore this hypothesis cannot be supported. This indicates that the relationship between cognitive existence and student satisfaction is not significant in this study. However, the path coefficients between satisfaction and teaching

**Table 7. Path coefficient test of structural equation model.**

| Path | | | Estimate | S.E. | C.R. | P Label | Std. Coef. | Conclusion |
|------|---|---|----------|------|------|---------|------------|------------|
| Perceived Usefulness | <— | Perceived Playfulness | .198 | .061 | 3.230 | .001 | .20 | Supported |
| Perceived Usefulness | <— | Cognitive Presence | .287 | .065 | 4.397 | *** | .29 | Supported |
| Perceived Usefulness | <— | Teaching Presence | .240 | .057 | 4.228 | *** | .24 | Supported |
| Perceived Usefulness | <— | Social Presence | .073 | .043 | 1.707 | .088 | .07 | Not Supported |
| Student Satisfaction | <— | Cognitive Presence | .014 | .072 | .186 | .852 | .01 | Not Supported |
| Student Satisfaction | <— | Teaching Presence | .272 | .069 | 3.958 | *** | .27 | Supported |
| Student Satisfaction | <— | Social Presence | .094 | .046 | 2.037 | .042 | .09 | Supported |
| Student Satisfaction | <— | Perceive Course Quality | .261 | .064 | 4.060 | *** | .26 | Supported |
| Student Satisfaction | <— | Perceived Usefulness | .337 | .080 | 4.203 | *** | .34 | Supported |
| Learning Attitude | <— | Student Satisfaction | .433 | .063 | 6.895 | *** | .43 | Supported |
| Continuous Learning Intention | <— | Learning Attitude | .312 | .062 | 5.030 | *** | .31 | Supported |

$*p < 0.05$;

$**p < 0.01$;

$***p < 0.001$

presence, perceived quality, and perceived usefulness were 0.272, 0.261, and 0.337, respectively, all at a significant level ($p < 0.001$), supporting the corresponding hypothesis. Indicating that the existence of teaching, perceived course quality, and system usefulness have a significant positive impact on student satisfaction. Finally, the path coefficient between learning attitude and satisfaction is 0.433, with a significance level ($p < 0.001$), supporting the hypothesis. This means that learning attitude has a significant positive impact on student satisfaction, that is, a positive learning attitude of students helps to improve their satisfaction level. In summary, structural equation modeling path analysis provides us with detailed hypothesis testing results, revealing the roles and complex relationships of various factors in the research model. These findings not only validate the theoretical framework of the study, but also provide a deep theoretical basis for further understanding the influencing factors and their mechanisms of action.

**2) Analysis of learning style moderates the relationship between perceived usefulness and learning attitude.** In Table 8, this study can explore in detail the moderating effect of learning style on the relationship between perceived usefulness and learning attitude. Firstly, the estimated value of the constant term is 4.1020, with a standard error of 0.5090, indicating the significance of the intercept term in the model. The path coefficient of student satisfaction with learning attitude is −0.6763, with a standard error of 0.1416, which is significantly negatively correlated with learning attitude ($p < 0.001$). In addition, the coefficient of influence of learning style on learning attitude is −0.3861, with

**Table 8. The result of learning style moderates the relationship between perceived usefulness and learning attitude.**

| Experimental result | | | | |
|---|---|---|---|---|
| **Model path** | **Estimate** | **S.E.** | **C.R.** | **P value** |
| Constant | 4.1020 | .5090 | 8.0583 | .0000 |
| Student Satisfaction | −.6763 | .1416 | −4.7756 | .0000 |
| Learning Style | −.3861 | .1552 | −2.4882 | .0134 |
| Student Satisfaction × Learning Style | .2449 | .0401 | 6.1082 | .0000 |
| Conditional effects of regulatory effects | | | | |
| **Learning Style** | **Effect** | **S.E.** | **t** | **P value** | **LLCI** | **ULCI** |
| 2.2500 | −.1253 | .0636 | −1.9699 | .0497 | −.2505 | −.0001 |
| 4.0000 | .3032 | .0534 | 5.6813 | .0000 | .1982 | .4082 |
| 4.5000 | .4256 | .0657 | 6.4808 | .0000 | .2964 | .5549 |

a standard error of 0.1552, which is also significant (p = 0.0134), indicating that learning style has a negative impact on learning attitude. In terms of interaction, the interaction term between student satisfaction and learning style shows a significant moderating effect. The estimated coefficient of interaction effect is 0.2449, with a standard error of 0.0401, and the P value is significant (p < 0.001), indicating that learning style plays a significant moderating role between student satisfaction and learning attitude. Further conditional effects analysis shows that the moderating effect is significantly different when different values are taken for learning styles. Especially when the learning style value is 4.5, the moderating effect is most significant, indicating that a high learning style can significantly enhance the relationship between student satisfaction and learning attitude.

   3) **Analysis of learning style moderates the relationship between learning attitude and continuous learning intention.** According to the analysis results in Table 9, the estimated constant term in the model is 2.3596, with a standard error of 0.5896, indicating the statistical significance of the intercept term (p = 0.0001). The path coefficient of learning attitude on continuous learning intention is 0.0159, with a standard error of 0.1919, but its p-value is 0.9340, indicating that this path coefficient is not statistically significant. The coefficient of influence of learning style on sustained learning intention is 0.2553, with a standard error of 0.1742. Although the p-value is 0.1438, it has not reached a significant level (p > 0.05), indicating that the direct impact of learning style on sustained learning intention is not statistically significant. In terms of interaction, the interaction term between learning attitude and learning style shows a lower moderating effect. The estimated coefficient of the interaction effect is 0.0173, with a standard error of 0.0504, and the P value is not significant (p = 0.7312), indicating that the moderating effect of learning style on the relationship between learning attitude and sustained learning intention is weak or non-existent. The results as displayed in Table 10 and Fig. 4. In sum, these analysis results indicate that under the current research conditions, the moderating effect of learning style on the relationship between learning attitude and sustained learning intention is not significant. This discovery suggests that when designing and implementing educational strategies, it may be necessary to pay attention to other factors that may affect learning attitudes and sustained learning intentions, to understand individual differences more accurately in the learning process and the formation of learning motivation.

## 4.6 Hypothesis testing

   This study evaluated the effects of internet technology on college elective music courses using factor analysis and structural equation modeling. The integration of both quantitative and qualitative analyses fortified the study's scientific rigor and reliability, providing a robust theoretical foundation for understanding the practical implications of internet technology in music education. Analysis of Table 9 and Fig 4 revealed that the incorporation of online resources and interactive learning environments fostered more active participation among students. These findings underscored the positive relationship between learning attitudes and a willingness to engage in continuous learning. However, certain hypotheses were unsupported, notably the connection between students' perceptions of internet technology's usefulness and their satisfaction, underscoring the multifaceted nature of factors influencing student satisfaction. Further exploration of these complexities from various theoretical perspectives is warranted.

**Table 9. The result of Learning style moderates the relationship between learning attitude and continuous learning intention.**

| Experimental result | | | | |
|---|---|---|---|---|
| Model path | Estimate | S.E. | C.R. | P value |
| Constant | 2.3596 | .5896 | 4.0023 | .0001 |
| Learning Attitude | .0159 | .1919 | .0829 | .9340 |
| Learning Style | .2553 | .1742 | 1.4654 | .1438 |
| Learning Attitude × Learning Style | .0173 | .0504 | .3438 | .7312 |

**Table 10. Hypothesis Classification.**

| No. | Hypothesis | Remark |
|---|---|---|
| H1 | Learning attitude positively correlates with continuous learning intention. | Supported |
| H2a | Learning style moderates the relationship between perceived usefulness and learning attitude. | Supported |
| H2b | Learning style moderates the relationship between learning attitude and continuous learning intention. | Not Supported |
| H3 | Student satisfaction positively correlates with learning attitude. | Supported |
| H4 | Perceived usefulness positively correlates with student satisfaction. | Not Supported |
| H5 | Perceive course quality positively correlates with student satisfaction. | Supported |
| H6 | Social presence positively correlates with student satisfaction. | Supported |
| H7 | Social presence positively correlates with perceived usefulness. | Not Supported |
| H8 | Teaching presence positively correlates with student satisfaction. | Supported |
| H9 | Teaching presence positively correlates with perceived usefulness. | Supported |
| H10 | Cognitive presence positively correlates with student satisfaction. | Not Supported |
| H11 | Cognitive presence positively correlates with perceived usefulness. | Supported |
| H12 | Perceived playfulness positively correlates with perceived usefulness. | Supported |

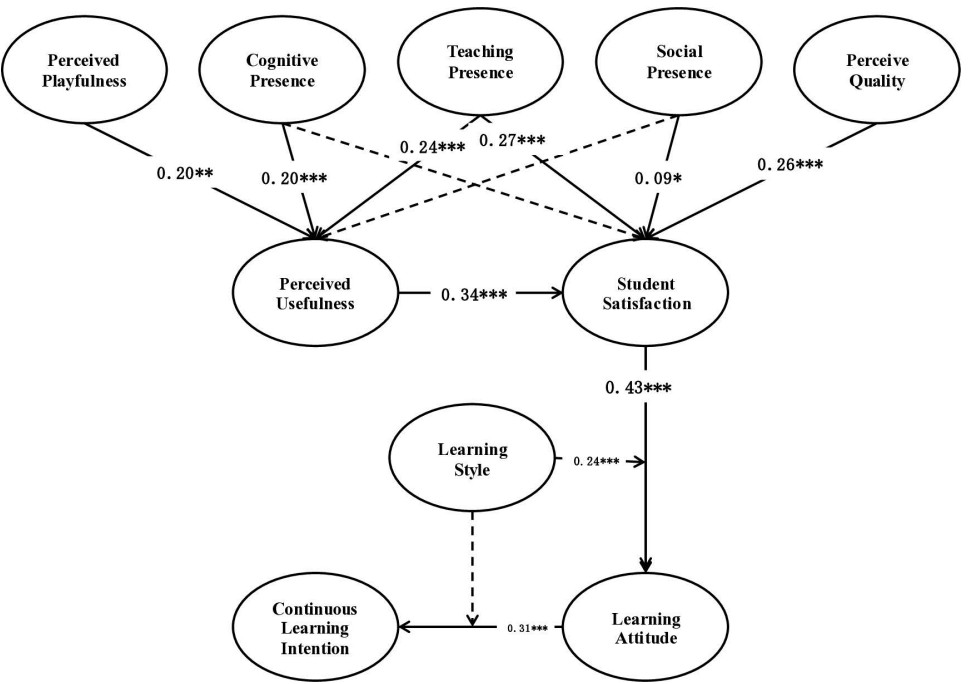

**Fig 4. Hypothesis Path Coefficient.**

Our findings indicate a significant positive correlation between student satisfaction and learning attitudes, suggesting a positive influence of internet technology on students' learning attitudes. Additionally, there is a notable correlation between perceptions of course quality and overall learning satisfaction, emphasizing the importance of course content design, teaching quality, and teaching methods. Social presence significantly correlates with students' overall satisfaction, highlighting the significance of social interaction in enriching course experiences. While the relationship between social presence and perceptions of internet technology's usefulness was not confirmed, this may stem from the dynamics of

social interactions. The critical role of teachers is underscored by the significant positive correlation between teaching presence and overall student satisfaction. A positive teaching presence allows teachers to increase students' interest in learning and enhance their evaluations and satisfaction with the course. This indicates that teacher involvement and effective teaching strategies are crucial in enhancing students' perceptions and experiences with online learning tools. The significant positive correlation between cognitive presence and students' perceptions of internet technology's usefulness highlights the importance of cognitive engagement in effectively using learning tools in online environments. Moreover, cognitive engagement plays a pivotal role in effectively utilizing learning tools in online environments, as indicated by the significant positive correlation between cognitive presence and students' perceptions of internet technology's usefulness. Overall, this study provides theoretical support for current music education practices and suggests avenues for future educational technology applications and developments. The findings underscore the potential of internet technology in music education while also highlighting the need for further research to deepen our understanding of its impacts across diverse learning environments and student groups."

## 5. Implications

### 5.1 Theoretical implications

The traditional education model is facing transformation. How to effectively use the Internet and multimedia technology and innovate music teaching design has become the key to enhance students' learning intention and effectiveness. Students can access a wide range of music resources and interact with music masters through online platforms, enriching their learning experience and enhancing their interest in learning through blended learning methods. Interesting course design can significantly improve students' learning satisfaction and willingness to continue learning. Introducing multimedia technology and interactive teaching activities can effectively stimulate students' interest in learning, enabling them to gain a deeper experience and sense of achievement in music learning. Meanwhile, blended learning not only expands the boundaries of traditional music theory teaching, but also promotes students' application ability in practical music performance. In today's rapidly developing information technology, the education sector is undergoing profound changes. The use of multimedia technology to enhance classroom teaching provides a solid material foundation for better education in schools [66]. By integrating and optimizing different teaching resources, the maximization of teaching effectiveness is achieved, thereby promoting students' continuous learning intention for music elective courses. The theory of sustained intention emphasizes the degree of learners' inclination towards learning behaviour over a longer period, especially in online learning environments, where social and instructional interactions are of great significance in enhancing sustained learning intention [67]. Interactive instructional design can not only reduce learners' anxiety and loneliness, but also improve their learning motivation, thereby enhancing the attractiveness and practical application ability of music elective courses [68]. The innovative teaching design of music elective courses needs to focus on the practicality of teaching content and the richness of learning experience. The integration of Internet and multimedia technology has brought unprecedented opportunities for the development of music education in colleges and universities, but educators need to continue to explore and experiment to better meet students' learning needs and expectations.

### 5.2 Practical implications

The teaching philosophy and technological innovation of college music elective courses are implemented in practical teaching practice. Firstly, multimedia technology and interactive teaching activities have been proven to effectively enhance students' interest and satisfaction with learning [69]. By introducing these teaching strategies, a more attractive and interactive learning environment can be created, stimulating students' enthusiasm and deep participation in music learning [70]. In addition, the theory of sustained intention emphasizes the importance of social interaction and instructional interaction in the online learning environment, which is crucial for enhancing students' learning motivation and sustained learning intention [71]. Therefore, educators should make full use of online platforms in design practice, establish

diverse learning interaction and feedback mechanisms to promote communication and learning achievement among students. In summary, by combining advanced educational technology and theoretical foundations, we can achieve more effective teaching design for music elective courses, enhance students' learning experience and practical application abilities, and thus promote the development and innovation of higher education music education.

## 6. Conclusion

The rapid development of the Internet has presented both opportunities and challenges for teaching music elective courses in higher education. This study delves into the application of blended learning models in music education, emphasizing the influence of students' learning attitudes and teaching methodologies. By integrating online learning with traditional classroom instruction, the blended learning model offers students more adaptable and personalized learning pathways. The online platforms facilitate students' exploration of additional music resources and interactions with teachers and peers, thereby fostering a stronger sense of engagement and motivation. Our research reveals associations between various themes, metrics, and students' learning attitudes. Specifically, understanding students' individual learning preferences through surveys or pre-tests can inform teaching strategies tailored to different learning styles, potentially enhancing students' interest and satisfaction. Furthermore, the study highlights the positive impact of interactive teaching, as measured by social presence, cognitive presence, and teaching presence, on students' learning experiences. Factors such as learning attitudes and styles play crucial roles in students' willingness to continue their music elective courses. Theoretically, this study incorporates the Technology Acceptance Model (TAM) to explore the mediating effects of perceived fun and perceived quality on students continued learning intentions. The findings indicate that students' perceived course quality and interest directly influence their learning satisfaction and attitudes, ultimately shaping their willingness to continue. Practically, the study advocates for the use of multimedia technology to enrich classroom instruction, optimize learning experiences, and bolster teaching effectiveness. For instance, introducing virtual reality (VR) technology can immerse students in diverse musical contexts, fostering creativity and broadening their musical horizons. Additionally, timely post-class feedback allows teachers to adjust their strategies based on students' progress. In summary, this study contributes both theoretically and practically to the understanding and promotion of innovative teaching practices in music elective courses. However, it is important to note the limitations of this study, including the potential lack of sample representativeness and biases in data collection and analysis. Future research should address these limitations by expanding the sample range and employing more rigorous methodologies to ensure the generalizability and objectivity of the findings.

## Supporting information

**S1 Appendix. Questionnaire.**
(PDF)

**S2 Data.**
(XLSX)

## Author contributions

**Conceptualization:** Xiaoshi Zhou.

**Formal analysis:** Pingbo Tang, Xiaoshi Zhou, Yuefeng Zhou, Rong Cheng.

**Funding acquisition:** Xiaoshi Zhou, Rong Cheng.

**Investigation:** Xiaoshi Zhou, Rong Cheng.

**Methodology:** Rong Cheng.

**Supervision:** Yuefeng Zhou.

**Writing – original draft:** Pingbo Tang.

**Writing – review & editing:** Pingbo Tang, Yuefeng Zhou.

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
