## [Decision Letter · Decision Letter 0]

PONE-D-24-33225How Do Multimedia and Blended Learning Enhance Music Elective Courses? Examining the Roles of Learning Attitudes, Styles, and Teaching PresencePLOS ONE

Dear Dr. Tang,

Thank you for submitting your manuscript to PLOS ONE. After careful consideration, we feel that it has merit but does not fully meet PLOS ONE’s publication criteria as it currently stands. Therefore, we invite you to submit a revised version of the manuscript that addresses the points raised during the review process.

We look forward to receiving your revised manuscript.

Kind regards,

Levent ÇALLI, Ph.D

Academic Editor

PLOS ONE

Additional Editor Comments (if provided):

Reviewers' comments:

Reviewer's Responses to Questions

**Comments to the Author**

1. Is the manuscript technically sound, and do the data support the conclusions?

Reviewer #1: Partly

Reviewer #2: Yes

2. Has the statistical analysis been performed appropriately and rigorously? 

Reviewer #1: No

Reviewer #2: Yes

3. Have the authors made all data underlying the findings in their manuscript fully available?

Reviewer #1: No

Reviewer #2: No

4. Is the manuscript presented in an intelligible fashion and written in standard English?

Reviewer #1: Yes

Reviewer #2: Yes

5. Review Comments to the Author

Reviewer #1: This study aims to identify interactions and causalities between the factors regarding e-learning in music elective courses. I think this study has shortcomings. My recommendations are listed below.

1. Remove the title “3.10. Conceptual Model.” Instead of this, a short explanation of the conceptual model should be given after the title “3. Hypothesis." And Figure 1 should be given in this part.

2. The literature shows that the factor "trust" was integrated with TAM. Why didn’t you examine the trust factor's relationship and influence?

3. The population of research is not given in the study. If you adopted the sampling, how did you select the sample? What is the sampling method? What is the size of the sample? If you targeted a limited population, what are the characteristics of this target population? And what is the response rate of participants? The answers you give show us the power and validity of your research.

4. I noticed you applied SEM in your research. But you did not employ the Confirmatory Factor Analysis (i.e. the Measurement Model), and the exploratory factor analysis was used. Why did you employ this approach?

5. “This study explored the potential factor structure in measurement tools using principal component analysis (PCA) and variance maximization rotation methods.” You should give reasons why you selected these methods. For example, the maximum likelihood method and the oblique rotation method can be used.

6. As I understand it, you used Eigenvalue 1 to determine the factors. Why did you employ this approach?

7. “Second, according to indicators such as NFI, RFI, IFI, TLI, and CFI, all exceeded the standard value of 0.9.” As evidenced by this phrase, you stated that all goodness of fit test results for your model ensure the standards. However, when I looked at Table 4, I noticed that NFI, GFI and RFI were 0.87. These values did not meet the requirements for goodness of fit.

8. You cite research that include judgment standards for goodness of fit tests and reliability and validity analyses.

9. The term “correlation” in your hypothesis is not correct because you applied SEM in your study.

10. Add a discussion section before the implications. The results obtained in this study should be discussed in depth with the studies on the subject in the literature.

Reviewer #2: This study provides valuable insights into the effects of multimedia and blended learning within music elective courses. Although this study contributes meaningfully to both theoretical and practical fields, I suggest that author strengthen the academic rigor and relevance of your paper by elaborating on and refining several critical elements.

1.Introduction

1)The lines 6-7 of the first paragraph mentions, “According to the guidelines of the Chinese Ministry of Education for aesthetic education in higher education institutions.”. It is essential to provide the publication year, thereby facilitating readers in locating and verifying the source of this information.

2)The lines 8-9 of the first paragraph mentions, “music elective courses in universities have always attracted numerous students with their unique charm.”. Are there particular data sets or research studies that support this assertion? For example, what proportion of students participates in music courses, and how do these figures contrast with those of other elective offerings?

3)The last sentence of the first paragraph mentions, “in the fast-paced modern society, to stimulate students’ sustained interest in music elective courses has become an interesting research context that is worth further investigation.”. This could be elaborated by discussing why this is an interesting research context, and how such sustained interest could potentially impact students' long-term learning outcomes and psychological well-being.

4)The line 1 of the second paragraph mentions, “learning intention (LI) has been studied extensively”, however, it does not provide a clear definition of the term or elucidate its significance within the specified context. The authors are encouraged to include more detailed definitions or contextual information within the manuscript.

5)The line 3-5 of the second paragraph mentions, “Music elective course teaching has been innovated in recent years by Internet and teaching methods to effectively meet the needs of students’ physical and mental development.” However, the manuscript does not provide specific information regarding the mechanisms through which these innovations have been enacted. For instance, it remains unclear whether these advancements are supported by an online platform, or specific interactive technologies. Additionally, while the manuscript asserts that “effectively meet the needs of students' physical and mental development.”. But, it would be advantageous to include concrete examples or evidence that demonstrate the effectiveness of these new approaches in meeting those needs. For instance, is there empirical data available concerning student feedback, improved learning outcomes, or enhancements in mental health?

6)The second paragraph mentions, the manuscript transitions directly from discussing learning intentions to statements about innovating music elective courses through the internet and teaching methods, without a bridge linking these two concepts. There is a lack of explanation on how the internet and new teaching methods respond to or promote the development of learning intentions. It is recommended that the author consider adding some explanations or data to support this connection.

7)The line 5-7 of the second paragraph mentions, “Even though multiple studies have shown that specific elective courses can improve students' performance in each subject area, and some studies have also evaluated student satisfaction at the end of the course, ”, please cite the relevant literature.

8)The line 7-9 of the second paragraph mentions, “there has not been a study conducted to determine students' expectations and continuous learning intention before taking music elective courses.”. Nevertheless, this assertion requires substantiation through a comprehensive review of the existing literature to establish the existence of this research gap.

9)The line 9-10 of the second paragraph mentions, “The preference of college students for the teaching methods of music elective courses has not been evaluated.”. Please cite the relevant literature.

10)The line 5-6 of the third paragraph mentions, “Maximizing teaching effectiveness via the integration and optimization of different teaching resources.”. This statement necessitates a grammatical adjustment.

11)The line 8-15 of the third paragraph mentions, the text addresses both practical perspective and theoretical perspective�however, it does not provide explicit methodologies for their integration. For example, the text does not clarify how to effectively merge the teaching of traditional music theory with practical performance. The incorporation of examples or comprehensive descriptions of specific pedagogical strategies would significantly improve the clarity and applicability of the discourse.

12)The last sentence of the third paragraph mentions, “This study further promotes the formation of a detailed mechanism for the relationship between the above factors.”. This formulation is relatively abstract. It is advisable to include specific information concerning the methodologies utilized to investigate or substantiate these mechanisms.

2.Theoretical background

13)The literature review section primarily highlights positive outcomes. Has there been any deliberation regarding the inclusion of opposing viewpoints or negative instances to augment the thoroughness and depth of the manuscript?

2.1

14)The lines 2 of the first paragraph mentions, Bhattacharjee’s concept is mentioned�please cite the relevant literature.

15)The lines 5 of the first paragraph mentions, the text experiences a sudden shift from theoretical to information systems. To improve the coherence of the discourse, it would be advantageous to incorporate one or two transitional sentences.

16)Literature cited in manuscript by Ng, Davy TK and Yao et al.

2.2

17)The first paragraph mentions,the TAM and its two fundamental components, but the ECM-ISC is suddenly introduced later without a clear explanation of the relationship between these two models.This deficiency in clarity may result in confusion for the reader, as the justification for the introduction of the ECM-ISC model, along with the referenced literature, is not clearly articulated. It is advisable to include transitional sentences to improve the logical coherence between the two models.

3. Hypotheses

3.1

18)The term “Learning attitude” is defined at the beginning of the paragraph, but subsequently, it is used interchangeably with “Learning motivation”, which may lead to conceptual confusion. It is recommended to clearly differentiate these two concepts or explicitly clarify their relationship.

19)Could author supply further academic literature that specifically examines the correlation between “Learning attitude” and “continuous learning intention”, in order to strengthen the scholarly foundation for the hypothesis?

3.3

20)Hypothesis H3 “Student satisfaction positively correlates with learning attitude.”. The preceding discourse predominantly highlights the connection between learning satisfaction and continued learning. It would be prudent to incorporate a discussion regarding the impact of satisfaction on attitudes toward learning before articulating the hypothesis.

3.4

21)Line 8: citation format needs to be standardized

3.5

22)“Perceive quality” should be corrected to “Perceived quality” throughout the document to maintain consistency and accuracy.

3.6

23)The line 14 of the paragraph mentions, “Hu confirmed that social presence is an important predictor of online learner satisfaction.”. Please cite the relevant literature.

3.7

24)The text defines “Teaching presence refers to students’ perception of the existence of teachers.”. However, this definition is somewhat succinct and may prompt readers to inquire whether it specifically addresses the physical presence of the teacher or the teacher’s engagement and participation in the educational process. It is advisable to provide a more detailed and nuanced definition.

25)Suppose there is a small error in H8: an extra “e”.

3.9

26)Please cite the relevant literature for the first sentence

4. Experimental results

4.1

27)The paragraph does not mention how the data were collected or the representativeness of the sample. Additionally, it should be clarified which methods were used for data analysis.

4.2

28)Ensure that all textual statements are directly supported by the data presented in the figures. As shown in the figure, students aged 18-20 have a higher proportion of choosing “agree,” “do not care,” “strongly disagree,” and “disagree.” Students aged 21-23 are more likely to choose “strongly agree.” Female students have higher proportions of choosing “strongly agree,” “do not care,” and “strongly disagree” compared to male students. The highest proportion of choosing “agree” is among juniors and freshmen, while the highest proportion of choosing “strongly agree” is among juniors and seniors.

4.3

29)If possible, please provide the findings of the Bartlett test.

4.4

30)The text indicates that the GFI is 0.877 < 0.9. This presents a logical inconsistency, as 0.877 is less than 0.9.

4.5

31)For the hypotheses that were not supported, could the author provide explanations or theoretical discussions? This is crucial for understanding the complexity and depth of the research findings.

32)There is no table 9 and the serial numbers need to be changed.

5. Implications

5.2

33)The line 4 of the paragraph paying attention to the correct use of punctuation.

6. Conclusion

34)What particular recommendations does the research offer regarding the design and execution of music elective courses in practice? Furthermore, can practical pedagogical strategies be suggested based on the study’s findings?

35)The study doesn’t adequately discuss the limitations of the study, particularly regarding the representativeness of the sample in relation to the wider student population in China. Furthermore, it is essential to evaluate whether the data collection and analysis methods utilized may introduce potential biases.

36)Kindly adjust the manuscript format to conform to the specifications outlined by PLOS ONE.

6. PLOS authors have the option to publish the peer review history of their article (what does this mean? ). If published, this will include your full peer review and any attached files.

**Do you want your identity to be public for this peer review?** For information about this choice, including consent withdrawal, please see our Privacy Policy .

Reviewer #1: No

Reviewer #2: No

---

## [Decision Letter · Decision Letter 1]

PONE-D-24-33225R1How Do Multimedia and Blended Learning Enhance Music Elective Courses? Examining the Roles of Learning Attitudes, Styles, and Teaching PresencePLOS ONE

Dear Dr. Tang,

Thank you for submitting your manuscript to PLOS ONE. After careful consideration, we feel that it has merit but does not fully meet PLOS ONE’s publication criteria as it currently stands. Therefore, we invite you to submit a revised version of the manuscript that addresses the points raised during the review process.

We look forward to receiving your revised manuscript.

Kind regards,

Levent ÇALLI, Ph.D

Academic Editor

PLOS ONE

Reviewers' comments:

Reviewer's Responses to Questions

**Comments to the Author**

1. If the authors have adequately addressed your comments raised in a previous round of review and you feel that this manuscript is now acceptable for publication, you may indicate that here to bypass the “Comments to the Author” section, enter your conflict of interest statement in the “Confidential to Editor” section, and submit your "Accept" recommendation.

Reviewer #1: (No Response)

Reviewer #3: (No Response)

2. Is the manuscript technically sound, and do the data support the conclusions?

Reviewer #1: Partly

Reviewer #3: Partly

3. Has the statistical analysis been performed appropriately and rigorously? 

Reviewer #1: No

Reviewer #3: (No Response)

4. Have the authors made all data underlying the findings in their manuscript fully available?

Reviewer #1: No

Reviewer #3: (No Response)

5. Is the manuscript presented in an intelligible fashion and written in standard English?

Reviewer #1: Yes

Reviewer #3: (No Response)

6. Review Comments to the Author

Reviewer #1: It has been observed that the authors have tried to revise the article in line with the criticisms and suggestions of reviewers. However, there are still some shortcomings.

1) Remove NFI, GFI, and RFI results from Table 4. Moreover, the comments related to these GoF tests should be also removed.

2) How to determine sample size in your study? As i understood, the numbers of participants was 315. Is this meet your predetermined sample size?

3) In SEM applications, both exploratory factor analysis (EFA) and SEM are not performed on the same data. Because, a second data collection process is required to verify the exploratory design. And for this purpose, a CFA should be used to test the measurement model. As far as I understand, in your study, both EFA and SEM were performed with the same data. If there is a scientific basis to this, please indicate it in the study by citing the relevant scientific sources.

Reviewer #3: This article presents a study on the relationships between students’ learning attitudes and different factors in music elective courses. I have a few issues, largely with regards to the clarity of the presentation of the research and the interpretations therein. Firstly, a few general comments:

1. There are several points of inconsistency, namely:

a. Reference formatting. Style changes between sections but never in Vancouver style as stipulated in the PLOS submission guidelines. In-text citations seem to be missing in the bibliography and/or vice-versa, for example Guardiola et al., 2019 and Rodríguez et al., 2019, respectively, unless this is a typographical error? In any case, this should be checked carefully.

b. Plot and table formatting. These change throughout the document. In Figure 3 especially the text is quite small and hard to read.

c. Statistical reporting. Sometimes significance is referred to with p< statements, others precise p values are given. I don’t think using asterisks in text is necessary.

d. Writing quality. Some sections read quite clearly whereas others are quite difficult with long sentences and often statements that are rephrased and repeated. For example, in the introduction: “According to the "2022 National College Students' Elective Course Survey Report," college students show the highest preference for art elective courses, accounting for 48.3% [https://www.thepaper.cn/newsDetail_forward_17337059]. The survey found that college students have the highest preference for art elective courses, accounting for 48.3%, while they have the lowest preference for economics and management courses, accounting for only 7.1% [https://news.qq.com/rain/a/20220326A05XK700].” The writing throughout could be made more concise to help the reader follow the article. Proofreading should treat the article as one whole not in segmented parts. Further, check the appropriate referencing format for citing websites.

2. Consider how the article is structured. There is no explicit methods section; details are given in the Experimental Results section, but it would help the flow of the article and for the reader to understand what has been done and how it was analysed (see below) if there were a separate detailed methods (and analysis) description before reporting the results. It is a bit difficult to follow as it currently is.

3. Throughout the article you make statements that are unsubstantiated and should be backed up with references, for example in the Introduction “It can not only cultivate students' musical literacy and enhance their aesthetic ability, but also provide them with a channel to express emotions and release stress” / “Sustained interest can not only enhance students' learning motivation but also improve their mental health. Furthermore, continuous music learning can help students build a deeper artistic literacy and enhance their emotional expression skills”; in the Experimental Results section “This may be because female students are more susceptible to emotional influences from music and are more inclined to enrich their spiritual lives through music elective courses”. What is your basis of these statements? Any assertions or declarative statements should be backed up with legitimate academic sources. If there are none, the authors should clearly state where this is an expression of opinion.

Some additional points by section:

1. Experimental results

a. The questionnaire items could be more clearly signposted. Table 1 tells you the sources but not the actual questions, and there is no direction in the article to guide the reader to the supplementary material. Additionally, it would be interesting to see the loading coefficients for each item from the PCA. This could be included in a summary table which, if not added to the main article, should include a clear note about where it can be found. This would help not only for the reader to understand the details of the study but also increase the thoroughness of the results presentation. Presumably the ten components derived from the PCA reflect the ten questionnaire categories/constructs? This is also not explicitly stated. If it is not the case, it would be interesting to know and discuss. Did the PCA produce a simple solution? Or was there any overlap of some items across components?

b. Check for language accuracy and/or typos. For example, the second sentence of the Distribution of Elective Courses section says, ‘Based on the results in Figure 2…’, but Figure 2 depicts a model and does not show any results.

c. I am not sure I agree with your interpretations of Figure 3. Firstly, there is quite an imbalance of numbers in some of the groupings, especially age. This should at the very least be acknowledged as a limitation when making any group-based comparisons. Also, is the title accurate? ‘Distribution of elective courses’, but the labels state ‘continuous learning intention’. You state that the 21-23 group were highest in the ‘agree’ and ‘strongly agree’ categories yet that’s only the case in the latter; the 18-20 group have a much higher count in ‘agree’. Unless you are referring to differences within that age group only, in which case that should be made clearer. You also say that students 18-20 have a higher proportion in the disagree categories; that appears to be true in terms of magnitude, but this plot isn’t showing the data proportionally, and as previously stated there appears to be an imbalance of numbers between those groupings. You also say that students aged 24-26 mainly focused on the ‘don’t care’ option, but that’s not what the plot shows either. Think carefully about how you are presenting your sample. You should explicitly state (or perhaps present in a table) a breakdown of your sample in terms of relevant demographics; it is not easy to glean those details from the figure presented. Demographic information should be included as a reporting necessity but be careful of what comparisons you make (if any) if your groupings are heavily skewed, or any suggestions of their implications. I would avoid making statements about ‘significant differences’ if you have not performed any statistical tests.

d. Consider the structuring of this section. It may help the reader if you begin with a description and explanation of all the planned statistical analysis before presenting the results of each, in a logical order. Tables should come after they are referred to in the text.

e. A statement already commented on in the previous review still needs addressing: “the GFI is 0.877, which also exceeds the standard value of 0.9”. I believe you are suggesting that you consider values in the region of 0.9 to be considered acceptable, but this has not been clearly articulated and the sentence is, nonetheless, incorrect.

2. Conclusion

a. I would consider your interpretation of the results. You state, “Analysis of Table 9 and Figure 4 indicates that the blended learning model utilizing internet technology significantly increased students' interest and engagement in music elective courses”, but to show an increase you would have to have compared between different forms of learning, or over time, which you have not. What your results appear to show, if I have understood correctly, is associations between different themes and metrics and students’ attitudes to learning. This is still a valuable insight but consider your terminology carefully. Similarly, “Through comprehensive analysis of quantitative and qualitative data, this study confirmed the effectiveness of internet technology in enhancing students' learning interest, increasing satisfaction, and promoting continuous learning”. The language used suggests a comparative assessment (i.e., internet technology enhanced/increased/promoted learning compared to something else), which is misleading.

I hope my comments are useful and constructive. My thanks for your hard work and efforts.

7. PLOS authors have the option to publish the peer review history of their article (what does this mean? ). If published, this will include your full peer review and any attached files.

**Do you want your identity to be public for this peer review?** For information about this choice, including consent withdrawal, please see our Privacy Policy .

Reviewer #1: No

Reviewer #3: No

---

## [Author Response · Author response to Decision Letter 2]

22 Feb 2025

Dear Editor and Reviewers,

We sincerely appreciate the time and effort you have dedicated to reviewing our manuscript titled “How Do Multimedia and Blended Learning Enhance Music Elective Courses? Examining the Roles of Learning Attitudes, Styles, and Teaching Presence”. This study aims to explore the possibility of using Internet resources to enhance the educational effect of music elective courses in colleges and universities. The research results are of great significance for optimizing the design and teaching methods of music elective courses, providing theoretical support and empirical basis for promoting innovation in music education. We are grateful for the constructive feedback provided by Reviewer #1 and Reviewer #3, which has helped us improve the quality of our work. Below, we address the reviewer’s comments point by point and outline the revisions we have made to the manuscript. We believe that these revisions have significantly strengthened the manuscript and addressed the concerns raised by the reviewer. We are confident that the revised version meets the high standards of PLOS ONE and hope that it is now suitable for publication.

Thank you once again for your valuable feedback. We look forward to hearing from you regarding the next steps in the review process.

Sincerely,

Pingbo Tang

Hunan Vocational College of Science and Technology

---

## [Decision Letter · Decision Letter 2]

PONE-D-24-33225R2How Do Multimedia and Blended Learning Enhance Music Elective Courses? Examining the Roles of Learning Attitudes, Styles, and Teaching PresencePLOS ONE

Dear Dr. Tang,

Thank you for submitting your manuscript to PLOS ONE. After careful consideration, we feel that it has merit but does not fully meet PLOS ONE’s publication criteria as it currently stands. Therefore, we invite you to submit a revised version of the manuscript that addresses the points raised during the review process.

Dear Authors, Thank you for your thorough revisions and for addressing the comments provided by the initial two reviewers with great care. Your revised manuscript reflects significant improvement and thoughtful engagement with the feedback.

Following the submission of the revised version, the manuscript was sent for an additional review by a third reviewer. While this reviewer raised only a few additional points, we believe that addressing these comments will further strengthen your article.

At this stage, we are requesting minor revisions. Once you have made the necessary adjustments, we will proceed with the final steps toward acceptance.

Please let us know if you have any questions. We look forward to receiving your revised manuscript.

Best regards,

Dr. Gal Harpaz

We look forward to receiving your revised manuscript.

Kind regards,

Gal Harpaz, Ph.D.

Academic Editor

PLOS ONE

Journal Requirements:

Additional Editor Comments:

Dear Authors,

Thank you for your thorough revisions and for addressing the comments provided by the initial two reviewers with great care. Your revised manuscript reflects significant improvement and thoughtful engagement with the feedback.

Following the submission of the revised version, the manuscript was sent for an additional review by a third reviewer. While this reviewer raised only a few additional points, we believe that addressing these comments will further strengthen your article.

At this stage, we are requesting minor revisions. Once you have made the necessary adjustments, we will proceed with the final steps toward acceptance.

Please let us know if you have any questions. We look forward to receiving your revised manuscript.

Best regards,

Dr. Gal Harpaz

Reviewers' comments:

Reviewer's Responses to Questions

**Comments to the Author**

1. If the authors have adequately addressed your comments raised in a previous round of review and you feel that this manuscript is now acceptable for publication, you may indicate that here to bypass the “Comments to the Author” section, enter your conflict of interest statement in the “Confidential to Editor” section, and submit your "Accept" recommendation.

Reviewer #3: (No Response)

2. Is the manuscript technically sound, and do the data support the conclusions?

Reviewer #3: (No Response)

3. Has the statistical analysis been performed appropriately and rigorously? 

Reviewer #3: (No Response)

4. Have the authors made all data underlying the findings in their manuscript fully available?

Reviewer #3: (No Response)

5. Is the manuscript presented in an intelligible fashion and written in standard English?

Reviewer #3: (No Response)

6. Review Comments to the Author

Reviewer #3: Thank you very much for revising your article. The authors have clearly made significant amendments. In particular, the writing quality in terms of flow and readability has greatly improved. However, I still have a few qualms with the report.

The structure of the results section is still a bit meandering and confusing. I suggested a formal Analysis section; this would outline what procedures were followed and include any necessary justifications. For example, the explanation for applying EFA, CFA, and SEM on the same dataset, explained in section 4.5, would go there, before you report the results of these analyses. A strict Analysis section may not be necessary, but just as a way of thinking about how things are presented. At the moment, it is a bit odd to report half of the results, then give some explanation of your analysis method, before providing the rest of the results.

Some of the citations are a bit confusing. For example, reference to Song (2023) in the introduction, first paragraph, second to last sentence – the sentence makes a strong statement about music education; however, the cited article does not discuss music. Is this a speculative statement? Do you mean ‘music leaning activities may facilitate the development…’?

Check specifics of Vancouver referencing – in-text citations should be numbered.

Check statistical reporting standards, for example “χ2(df) = ” instead of “Chi Squared = / df = ”. Additionally, although you’ve removed the asterixis when reporting significance the sentences now look a bit odd, for example “the path coefficient of perceived usefulness on perceived playfulness is 0.198, with a significance level (p<0.001)”, why not “the path coefficient […] is 0.198, which was significant (p < .001)”, or “the path coefficient […] was significant (p < .001) at 0.198,”? (p in italics and removing the 0 before the decimal is commonly done)

Table formatting could still be improved, for example a description for Table 4 would be helpful – presumably the acronyms listed as each variable refer to the different measurement items, but some clarification/explicit statement and perhaps a reminder of what they are would be helpful. Table 6 (and others) – words breaking across lines.

Thank you for your continued work.

7. PLOS authors have the option to publish the peer review history of their article (what does this mean? ). If published, this will include your full peer review and any attached files.

**Do you want your identity to be public for this peer review?** For information about this choice, including consent withdrawal, please see our Privacy Policy .

Reviewer #3: No

---

## [Author Response · Author response to Decision Letter 3]

11 May 2025

Dear Reviewers,

We are deeply grateful for your thorough and constructive feedback on our manuscript, "How Do Multimedia and Blended Learning Enhance Music Elective Courses? Examining the Roles of Learning Attitudes, Styles, and Teaching Presence". Your insightful comments have been invaluable in helping us improve the quality and clarity of our work. We sincerely appreciate the time and effort you have dedicated to reviewing our paper, and we have carefully addressed each of your concerns in this revision. Your thoughtful comments have helped us significantly improve the quality of our work. We have carefully addressed each of your concerns and believe the revised manuscript is much stronger as a result. If you have any additional suggestions, we would be happy to consider them. Below, we provide a detailed response to your specific comments.

Sincerely

Pingbo Tang

Hunan Vocational College of Science and Technology

---

## [Editor Report · Decision Letter 3]

How Do Multimedia and Blended Learning Enhance Music Elective Courses? Examining the Roles of Learning Attitudes, Styles, and Teaching Presence

PONE-D-24-33225R3

Dear Dr. Tang,

We’re pleased to inform you that your manuscript has been judged scientifically suitable for publication and will be formally accepted for publication once it meets all outstanding technical requirements.

Kind regards,

Gal Harpaz, Ph.D.

Academic Editor

PLOS ONE

Additional Editor Comments (optional):

Dear Authors

I am pleased to inform you that the article has been accepted for publication, all the reviewers' comments were well received, and the article is certainly worthy of publication in the journal in its current version.
---

## [Editor Report · Acceptance letter]

PONE-D-24-33225R3

PLOS ONE

Dear Dr. Cheng,

I'm pleased to inform you that your manuscript has been deemed suitable for publication in PLOS ONE. Congratulations! Your manuscript is now being handed over to our production team.

Kind regards,

on behalf of

Dr. Gal Harpaz

Academic Editor

PLOS ONE